# A Precise Prediction Method for the Properties of API-Containing Tablets Based on Data from Placebo Tablets

**DOI:** 10.3390/pharmaceutics12070601

**Published:** 2020-06-28

**Authors:** Yoshihiro Hayashi, Kaede Shirotori, Atsushi Kosugi, Shungo Kumada, Kok Hoong Leong, Kotaro Okada, Yoshinori Onuki

**Affiliations:** 1Formulation Development Department, Development and Planning Division, Nichi-Iko Pharmaceutical Co., Ltd., 205-1, Shimoumezawa Namerikawa-shi, Toyama 936-0857, Japan; a-kosugi@nichiiko.co.jp (A.K.); shungo.kumada@nichiiko.co.jp (S.K.); 2Laboratory of Pharmaceutical Technology, Graduate School of Medicine and Pharmaceutical Science for Research, University of Toyama, 2630 Sugitani, Toyama-shi, Toyama 930-0194, Japan; kaede.shirotori@gmail.com (K.S.); kokada@pha.u-toyama.ac.jp (K.O.); onuki@pha.u-toyama.ac.jp (Y.O.); 3Department of Pharmaceutical Chemistry, Faculty of Pharmacy, University of Malaya, Kuala Lumpur 50603, Malaysia; leongkh@um.edu.my

**Keywords:** tablet, response surface method, quality by design, tensile strength, disintegration time, database

## Abstract

We previously reported a novel method for the precise prediction of tablet properties (e.g., tensile strength (TS)) using a small number of experimental data. The key technique of this method is to compensate for the lack of experimental data by using data of placebo tablets collected in a database. This study provides further technical knowledge to discuss the usefulness of this prediction method. Placebo tablets consisting of microcrystalline cellulose, lactose, and cornstarch were prepared using the design of an experimental method, and their TS and disintegration time (DT) were measured. The response surfaces representing the relationship between the formulation and the tablet properties were then created. This study investigated tablets containing four different active pharmaceutical ingredients (APIs) with a drug load ranging from 20–60%. Overall, the TS of API-containing tablets could be precisely predicted by this method, while the prediction accuracy of the DT was much lower than that of the TS. These results suggested that the mode of action of APIs on the DT was more complicated than that on the TS. Our prediction method could be valuable for the development of tablet formulations.

## 1. Introduction

Tablet properties are affected substantially by the physicochemical properties of the loaded active pharmaceutical ingredients (APIs) as well as by formulation factors and process parameters [1,2]. For instance, the tensile strength (TS) and the disintegration time (DT) of a tablet, which are fundamental properties, are affected by particle size, moisture, and the bulk density of the APIs. These tablet properties are very important for the formulation design of commercial tablets. However, it is difficult to predict them depending on these factors quantitatively in pharmaceutical development because the mode of action of APIs on a tablet’s properties is complicated and is substantially influenced by the drug loading.

A better understanding of the mode of action of these factors on the pharmaceutical characteristics of the tablets became more important after the introduction of the “quality by design (QbD)” concept at the International Conference on Harmonization Q8 guideline in 2008 [3]. According to the QbD concept, quality must be built into the product; pharmaceutical researchers are required to perform formulation design based on a systematic, scientific, risk-based, holistic, and proactive approach. A better understanding of the relationships between factors and characteristics would be valuable for developing desirable pharmaceutical products and contribute to a reduction in product variability and defects, thereby enhancing product development, quality, and manufacturing efficiencies. The response surface method (RSM) is regarded as being a powerful tool to implement the QbD approach [4]. In general, response surfaces representing factors and characteristics were estimated by a regression equation such as a quadratic polynomial after obtaining experimental data based on a design of experiment (DoE) method. RSM can visualize the causal relationships between the factors and the pharmaceutical characteristics, which were used to identify crucial formulation factors from numerous potential factors [5,6,7,8,9], to optimize formulation [10,11] or process parameters [11,12,13], and to construct design spaces [14,15,16,17,18].

Against this background, the establishment of a prediction method of tablet properties appears to be promising, with the capacity of achieving a significant reduction in the time, resources, and costs involved in pharmaceutical tablet development. Several technical reports have addressed this subject [19,20,21]. In a previous study, we investigated the correlation between response surfaces of placebo- and API-containing tablets to fully understand the mode of action of APIs on the tablet properties [22]. In brief, placebo- and API-containing tablets were prepared according to DoE, and then their TS and DT were measured. By comparing the response surfaces of the relationships between the formulation factors and tablet properties, the effect of the loaded APIs on the tablet properties was better understood. As a result, particularly the TS of all API-containing tablets tested obviously correlated with those of the placebo tablets. Moreover, we demonstrated that the TS of the API-containing tablets could be predicted accurately from small experimental data sets of API-containing tablets utilizing placebo data. This is a great advantage for the pharmaceutical development of tablets, especially in the early stage, because this method allows characterization of the overall properties with a small number of experiments. In general, conventional DoE requires ten or more formulations to be investigated to derive this information. Many pharmaceutical studies have used RSM; however, the application of RSM described here is novel.

For the proof of concept, it is necessary to accumulate further technical knowledge concerning the relationships of response surfaces between placebo- and API-containing tablets. First, in the previous study, all experiments were conducted with a drug load of 10%, which was relatively low. To demonstrate the practical advantage of the method for designing pharmaceutical formulations, the present study tested API-containing tablets with drug loads ranging from 20% to 60%. This study provides an efficient prediction method to gain a comprehensive understanding of tablet properties.

## 2. Materials and Methods 

### 2.1. Materials

Lactose monohydrate (LAC, Pharmatose 100M; DFE Pharma, Goch, Germany), cornstarch (CS, Graflow M; Nippon Starch Chemical, Osaka, Japan), microcrystalline cellulose (MCC, Ceolus PH-101; Asahi Kasei Chemicals, Tokyo, Japan), and acetaminophen (ACE; Yamamoto Corporation, Osaka, Japan) were purchased from the indicated suppliers. Magnesium stearate (Mg-St), *o*-ethoxybenzamide (ETZ), nicotinic acid (NA), and pyridoxine hydrochloride (PYH) were purchased from Wako Pure Chemical Industries (Osaka, Japan). Physicochemical properties of model APIs (ACE, ETZ, NA or PYH) and excipients such as particle size, particle shape and solubility are summarized in Table 1.

### 2.2. Tablet Formulation

Model tablets composed of API, LAC, CS, MCC, and Mg-St were prepared using the direct compaction method. The designated load of the APIs ranged from 20% to 60%. LAC, CS, and MCC were selected as formulation factors. Based on a previous study, the composition of the formulation factors was determined according to an extreme vertices design (Figure 1). Namely, ten formulations, including duplicates of the centroid, were prepared for each drug load condition (Table 2). As for API-containing tablets, the amount of Mg-St was fixed at 0.60%.

The placebo formulation was limited to excipients without API. The placebo tablets used the formulation of the 10% API-containing tablet; thus, the Mg-St content was 0.67%. For instance, Recipe (Rp.) 1 of 10% API-containing tablet is composed of 20 mg API, 17.88 mg CS, 160.92 mg MCC, and 1.20 mg Mg-St. As for the placebo tablet, Rp. 1 is composed of 19.87 mg CS, 178.80 mg MCC, and 1. 33 mg Mg-St.

In brief, all APIs were sieved through a 355-μm mesh. The other ingredients were dried at 75 °C for 8 h. Except for the lubricant, the designated load of each powder was placed into a polyethylene bag and mixed for 2 min. Then, the lubricant was added to the mixture and mixed for 1 min. The resulting tablet powder (200 mg) was compressed at 240 MPa into a round-faced tablet, 8 mm in diameter and with a 12 mm radius of curvature, using an eccentric tablet machine (AUTOTAB-100W; Ichihashi-Seiki, Kyoto, Japan). The compression pressure was applied from the lower punch side. The punch speed was set at 380 mm/min according to the default value. Note that the tableting speed drops to 20% (76 mm/min) when the compression pressure reaches 30% (72 MPa).

### 2.3. Evaluation of Tablet Properties

The TS and DT were selected as the tablet properties for the model. They are important properties in the early development phase of direct compression formulations, because they are related to the tablet’s friability, tabletability, and dissolution properties.

The tablet hardness was determined using a tablet hardness tester (Portable Checker PC-30; Okada Seiko, Tokyo, Japan). The TS was calculated as:(1)$$\mathrm{TS}=\frac{2F}{\pi dt},$$
where *F* is the maximal diametrical crushing force, and *d* and *t* are the diameter and thickness of the tablet, respectively. The TS of each formulation was measured in triplicate.

The disintegration test was performed according to the Japanese Pharmacopoeia, 17th edition (JP17) disintegration test for tablets using a disintegration tester (NT-20H; Toyama Sangyo, Osaka, Japan). Purified water was used as a solvent at 37 °C. The DT was defined as the interval required for the complete disappearance of a tablet or its particles from the tester net. The DT was measured for three tablets of each formulation.

### 2.4. Construction and Comparison of the Response Surfaces for the Tablet Properties of Placebo- and API-Containing Tablets

The response surfaces representing the relationships between formulation variables and tablet properties (TS and DT) were obtained using the response surface method that incorporates multivariate spline interpolation (RSM-S) [23]. The values of the tablet properties of untested formulations were extracted from reading points on the response surfaces. Five hundred untested formulations were selected randomly. Afterwards, the correlation between the tablet properties of the placebo- and API-containing tablets for each formulation was evaluated using a scatterplot. A linear regression equation was applied to evaluate their relationships quantitatively.

### 2.5. Prediction of the Response Surfaces Using Regression Analyses and Data from Placebo Tablets

The TS and DT of API-containing tablets were predicted using the following equations:(2)$$F_{\mathrm{API}}\left(X_{1},\;X_{2},X_{3}\right)=af_{\mathrm{placebo}}\left(X_{1},\;X_{2},X_{3}\right)+b,$$
(3)$$F_{\mathrm{API}}\left(X_{1},\;X_{2},X_{3}\right)=a{\left\{f_{\mathrm{placebo}}\left(X_{1},\;X_{2},X_{3}\right)\right\}}^{2}+bf_{\mathrm{placebo}}\left(X_{1},\;X_{2},X_{3}\right)+c,$$
where $f_{\mathrm{placebo}}$ and $F_{\mathrm{API}}$ are the tablet properties (TS or DT) of the placebo- and API-containing tablets, respectively, and $X_{1},\;X_{2}$, and $X_{3}$ are the amounts of LAC, CS, and MCC, respectively. The parameters *a*, *b*, and *c* were obtained by plotting the TS or DT for the two and four types of formulations. As for the DT, quadratic and linear regressions were applied to the data obtained from the left and right halves of the response surface, respectively. The details of the method were reported in a previous study [22].

### 2.6. Evaluation of the Prediction Accuracy

The tablet properties of the API-containing tablet for the 500 model formulations were predicted by using data of the placebo tablet according to Equation (2) or (3), and then they were compared with those estimated from their own response surfaces. Two measures were selected for evaluating the prediction accuracy quantitatively—the determination coefficient (*R*^2^) and the mean absolute error (MAE). These two measures are defined as follows:(4)$$R^{2}=1-\frac{\sum_{i=1}^{n}{\left(y_{i}^{obs}-y_{i}^{prd}\right)}^{2}}{\sum_{i=1}^{n}{\left(y_{i}^{obs}-{\bar{y}}_{i}^{obs}\right)}^{2}},$$
(5)$$\mathrm{MAE}=\frac{\sum_{i=1}^{N}\left|y_{i}^{obs}-y_{i}^{prd}\right|}{n},$$
where $y_{i}^{obs}$ is the TS or DT value of API-containing tablets predicted using the placebo tablet database according to Equation (2) or (3), while and $y_{i}^{prd}$ is those derived from response surfaces of API-containing tablets. ${\bar{y}}_{i}^{obs}$ is the mean of the observed TS or DT values, and *n* is the number of samples. In this study, *R*^2^ and MAE were calculated from 500 data predicted according to the procedure mentioned above.

### 2.7. Computer Programs

DoE was performed using JMP Pro (v. 14.2; SAS Institute, Cary, NC, USA). The RSM-S was performed using dataNESIA (v. 3.2; Azbil Corporation, Tokyo, Japan).

## 3. Results and Discussion

### 3.1. Response Surfaces for the TS and DT of Placebo- and API-Containing Tablets

This study investigated the mode of action of the formulation factors on the tablet properties of different API-containing tablets using RSM. After measurement of the tablet properties of each API-containing tablet (see Appendix A), their response surfaces were created and compared to those of the placebo tablet.

The response surfaces for the TS of placebo- and API-containing tablets are shown in Figure 2. Overall, the TS of the placebo tablets appeared to be determined by the MCC content. A higher TS was observed in the formulations enriched with MCC, while a lower TS was observed in the formulations containing lower amounts of MCC. MCC is well known to be an excipient that has excellent tabletability. MCC offers an increase in the TS of tablets compared with LAC and pregelatinized starch [24]. The higher TS of MCC might be attributed to its higher plasticity, which would result in a larger bonding area and, therefore, stronger tablets [25,26]. The absolute values of the API-containing tablets decreased steadily with an increase in drug load; for instance, ACE-containing tablets in Rp. 5 had a TS of 2.45 MPa at a drug load of 20%, while at a 60% load the TS was decreased by 0.82 MPa. Although the absolute TS values varied according to the API content, this tendency was seen in almost all API-containing tablets, even when the drug load was increased to 60%. However, ETZ-containing tablets in Rp. 7 and 8 differed; their TS increased with an increase in drug load. Among the tested APIs, ETZ had the smallest particle size (Table 1). The small particle size may be a reason for the higher TS here. The TS of tablets is affected by several API properties, such as particle size [27], moisture content [28], and elastic recovery [29]. In our previous study [30], we established that one of the most crucial factors is the particle size. In general, there is a negative correlation between particle size and TS [31,32] because the compaction of smaller primary particles provides a larger total area for bonding than in the case of larger particles.

The TS is also affected by scale effects such as the compaction profile, compaction speed, and agitation in the feed frame [33]. Generally, there is a possibility that the TS might decrease at the production scale because the compression speed at the production scale is faster than at the lab scale. For instance, the TS of pregelatinized starch tablets is substantially affected by the compression speed, whereas LAC and MCC are independent [24]. Thus, the TS of formulations containing higher amounts of CS (Rp. 8) might decrease in industrial production. The effect of the agitation in the feed frame also should be considered in the scale-up because overmixing leads to a decrease in TS when the tablet contains Mg-St as a lubricant. To consider the effect of these scale-up factors on the response surface, further study is required.

Regarding the DT, the placebo tablets prepared according to Rp. 5 and 8 showed higher values than the rest (Figure 3). We assumed that these longer DT were caused by distinct mechanisms. Rp. 5 was composed of a large amount of MCC (90%) and was free of CS, which was used as a disintegrant in the model tablets. As observed from the results of the TS (Figure 2), MCC particles were able to bind tightly, resulting in solid tablets with a higher TS. A higher TS is probably accompanied by a longer DT. On the other hand, Rp. 8 consisted of CS (60%) and MCC (40%), and did not contain any LAC. We note that the placebo tablets prepared according to Rp. 8 had longer DT, but lower TS. LAC was the sole component that could dissolve in water in the model tablets. We thought that the lack of soluble components was due to the prolonged DT.

Changes in the absolute values of the DT depend on the API content. In the cases of ACE- and PYH-containing tablets, the DT tended to become shorter with an increase in drug load, while in the cases of ETZ- and NA-containing tablets, the DT tended to become longer. The DT of tablets is strongly affected by several API properties, such as particle size, solubility, particle shape, hygroscopicity, chemical structure, and density of particles [34]. ETZ and NA are less soluble than PYH. Therefore, the lower solubility might lead to a decrease in the wettability and rate of water penetration into the tablet, resulting in a longer DT. The solubility of ACE was similar to that of NA; however, the TS of an ACE-containing tablet was lower than that of an NA-containing tablet. A lower TS indicates low interparticulate bonding strength, leading to a more rapid disintegration of the tablet.

This study further compared the response surfaces of the DT between the placebo- and API-containing tablets. The response of the DT to the drug load was assumed to be distinct between Rp. 5 and 8. Overall, the DTs of Rp. 8 were higher even when a higher amount of API was loaded into the tablets. By contrast, in most cases of API-containing tablets, the DT of Rp. 5 became shorter with increasing drug load. We also noted that for some APIs such as ETZ and NA, the pattern of the response surfaces was changed substantially with the increase of the drug load. For instance, the longest DT of NA-containing tablets were observed from Rp. 8 at a low drug load ranging from 20% to 30%, while the formulation areas with the longest DT moved to Rp. 5 and 6 when the drug load rose to 40% or more. These behaviors were quite different from those of the TS—the TS monotonically decreased with an increase in drug load that was independent from the loaded API. These results suggest that the action of APIs on the DT was more complicated and intense compared with the TS.

### 3.2. Relationships between Response Surfaces of Tablet Properties for Placebo- and API-Containing Tablets

To investigate the relationships of tablet properties between placebo- and API-containing tablets in detail, scatterplots of their experimental data were created by random sampling of the tablet properties from the response surfaces. When these response surfaces associate with each other, the scatterplots are supposed to show some correlations including linear line, quadric curve and so on. On the other hand, when there are no correlations, data points are distributed in a disorderly fashion in a large area.

Figure 4a shows a scatterplot matrix for the TS of each formulation. Linear correlations were observed clearly in all cases, indicating that the changes in TS according to the formulation factors were the same in the placebo- and API-containing tablets. This result was consistent with our previous report [22]. We further obtained regression lines between the placebo- and API-containing tablets and then calculated their slopes. The relationship between the slope of the regression lines and the drug load is shown in Figure 4b. All the slopes were smaller than one. This means that, as far as the tested APIs are concerned, the API-containing tablets always showed a lower TS than the placebo tablets. Furthermore, except for NA, the slope decreased linearly with increasing drug load, indicating a decrease in the absolute value of TS with increasing drug load. As for the NA, when the API content was low (20–40%), the slope was similar or slightly increased with increasing drug load. By contrast, for high API content (40–60%), the slope substantially decreased with increasing drug load. Although further investigation is required to understand these results fully, the effect of NS on TS seems to be more complex than that of the other APIs tested. From these findings, we assumed that a precise prediction for the TS of API-containing tablets is feasible using the linear correlation between the placebo- and API-containing tablets.

The scatterplot matrices for DT are shown in Figure 5a. Except for the ETZ- and NA-containing tablets with a high drug load (50–60%), most scatterplots appeared to be two distinct lines. A similar finding was also observed in our previous study investigating API-containing tablets with a 10% drug load [22]. We note that the orange and blue points in the figure correspond to the data points of the formulations distributed in the right and left halves of the response surface—data points of the formulations around Rp. 5 are depicted in orange, while those around Rp. 8 are depicted in blue (Figure 5b). It was obvious that the two lines were clearly distinguished by the formulation areas, indicating that such a difference derived from different responses of the DT to the drug load. Namely, the DT of Rp. 5 was very sensitive to the drug load—the high DT of the placebo tablet completely disappeared after only 20% of the drug was loaded. This was significantly visible in the slopes of the most orange lines that were close to zero. By contrast, the DT of Rp. 8 remained at a high level even after 60% of the APIs were loaded. Thus, in most cases, the blue line lies above the orange line. We further note that in some cases, e.g., in 50–60% ETZ- and NA-containing tablets, the scatterplots were distributed in a disorderly fashion. This indicated that there was no obvious correlation of DT between placebo- and API-containing tablets. From these findings, the mode of action of APIs on the DT was much more complicated compared with that of TS.

CS was selected as the disintegrant in this study, and its disintegration mechanism is known to consist mainly of swelling [35]. By contrast, crospovidone and croscarmellose sodium are also known as super disintegrants, and their disintegration mechanism consists not only of swelling but also of wicking [35]. Previously, we confirmed that when tablets contained not CS but croscarmellose sodium, there was also a correlation between the response surface for the TS and DT of the placebo- and the API-containing tablet, respectively [22]. Although we did not evaluate tablets containing super disintegrants in this study, such tablets might show a correlation between the response surface for placebo- and API-containing tablets, even when the API load is high, because their disintegration mechanism is different from that of CS. We believe that further study is needed to clarify this issue.

### 3.3. Prediction of Response Surfaces for Tablet Properties of API-Containing Tablets

We predicted the TS and DT of API-containing tablets based on the findings obtained from the above experiments. Namely, we expected that the tablet properties of any API-containing tablet can be predicted precisely by combining a small number of experimental data with the response surfaces of the placebo tablet. First, a couple of API-containing tablets were prepared. Next, the relationship between the API-containing and placebo tablets shown in Figure 4a and Figure 5a was estimated by the linear or quadratic equation. Consequently, the regression lines were used to predict the tablet properties of the target API-containing tablets. As far as the DT prediction was concerned, the response surface of the DT of the placebo tablet was divided into two formulation areas (orange and blue in Figure 5b). Thus, two distinct regression lines were created, and then one of them was used for the prediction. The estimated parameters of regression lines are summarized in Appendix A.

The prediction accuracy of the estimated responses surfaces was evaluated by *R*^2^ and MAE. Regarding the TS, every regression model was created by combining only two experimental data points of the tested API-containing tablets (Rp. 5 and 8) with the response surface of the placebo tablet. Figure 6 shows the *R*^2^ and MAE of the TS. As shown, *R*^2^ was higher than 0.9, while MAE was lower than 0.20 MPa in all cases, indicating a good estimation. Because MAE means the average of the difference between the actual and the predicted value, the prediction error was considered to be sufficiently low. Thus, this method is effective for the TS prediction of any API-containing tablet. As for the DT prediction, the data points of the API-containing tablets of Rps. 3, 4, 5, and 8 were used to create regression models; the quadratic regressions for the prediction of the left half of the response surface were created using the data points of Rps. 3, 4, and 8, and the linear regressions for the right half were produced using the data points of Rps. 4 and 5. The overall accuracy of the DT prediction was much lower than that of the TS—i.e., *R*^2^ and MAE varied significantly according to the amount and types of the API added (Figure 7). PYH-containing tablets showed a better *R*^2^ and MAE compared with the other tablets—i.e., *R*^2^ was higher than 0.6, while MAE was lower than 20 s with any drug load. In addition, the prediction accuracy for the DT decreased with increasing drug load. This indicates that the response surfaces of API-containing tablets became more different with increasing drug load.

Therefore, the prediction accuracy for the TS was high enough even if API-containing tablets were tested. It is known that numerous API properties have a significant impact on the TS. They include particle size, bulk density, moisture content, and others [30]. However, these effects are thought to be independent from the effects of the formulation factors of the model tablet, and a high prediction accuracy could be achieved. By contrast, the prediction accuracy for the DT was affected substantially by the amount and types of API that were loaded. DT is also affected by numerous API properties, e.g., specific surface area, density of the area, particle shape and hygroscopicity, and solubility [34]. Our results suggested that the contribution of API properties to the DT was more complicated than that to TS. To clarify this issue and to achieve a precise prediction of the DT, a larger dataset which contains many API varieties might be needed.

## 4. Conclusions

This study offers a novel method to accurately predict tablet properties of API-containing tablets. The key technique of the method is to compensate for the lack of experimental data using data of placebo tablets. To achieve a precise prediction, it is important to evaluate the relationships between the response surface of the placebo- and the API-containing tablets because the prediction accuracy depends on the correlation of the response surfaces. As for the TS, we found that the method enables an accurate prediction of the TS even with high drug loads (at least up to 60%). By contrast, the prediction accuracy of the DT was much lower than that of the TS, indicating that the mode of action of APIs on the DT was more complicated than that on TS. This methodology will help to design tablet formulations rapidly and effectively.

## Figures and Tables

**Figure 1 pharmaceutics-12-00601-f001:**
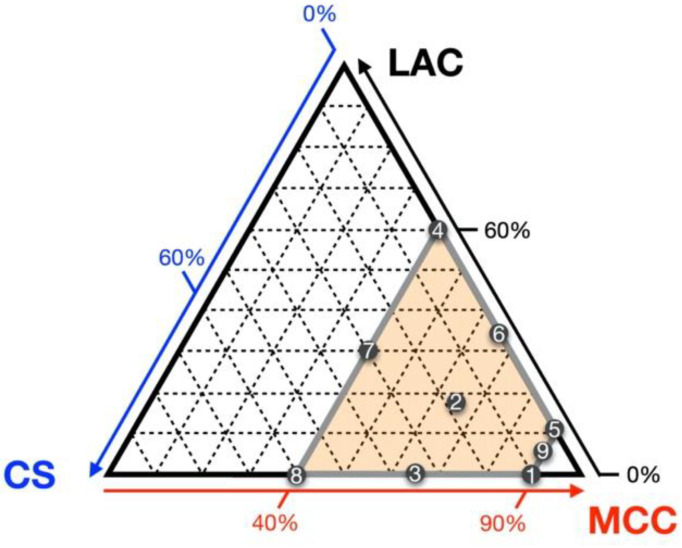
Excipient composition of model tablets. Each point represents the excipient composition and formulation number of one model tablet.

**Figure 2 pharmaceutics-12-00601-f002:**
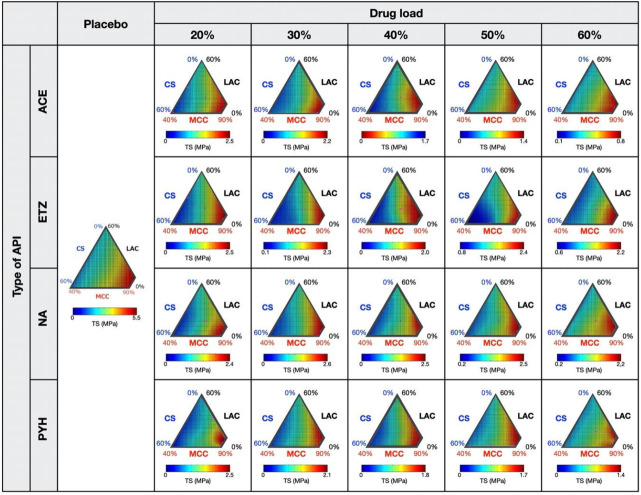
The response surface generated by multivariate spline interpolation (RSM-S) for the tensile strength (TS) of each tablet. The response surface of the placebo tablet is taken from our previous study [22].

**Figure 3 pharmaceutics-12-00601-f003:**
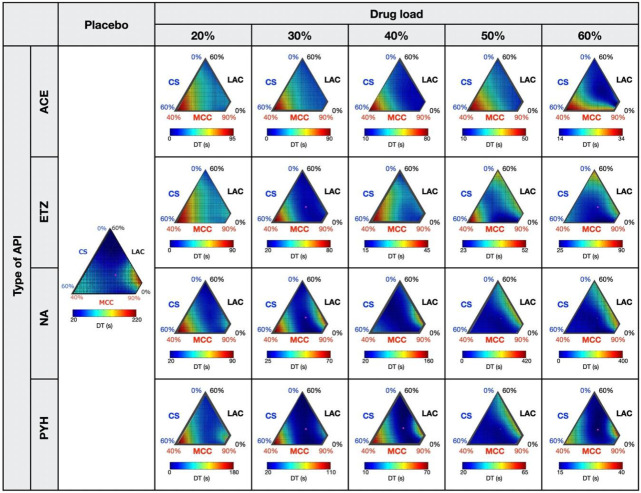
The response surface generated by RSM-S for the disintegration time (DT) of each tablet. The response surface of the placebo tablet is taken from our previous study [22].

**Figure 4 pharmaceutics-12-00601-f004:**
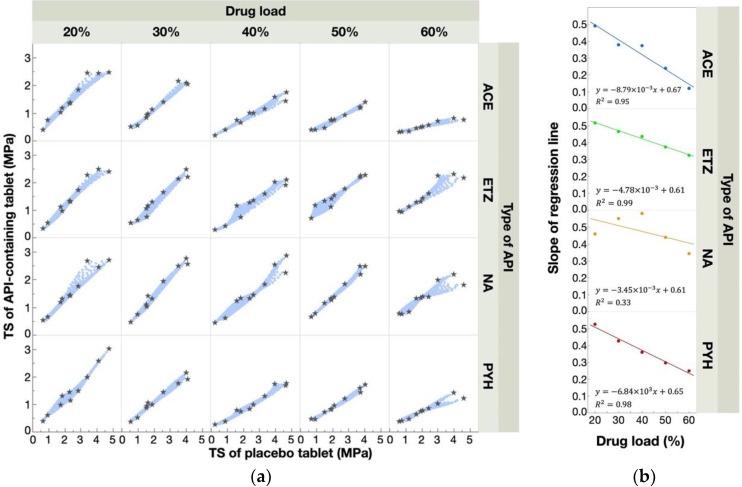
Relationships between the response surface for the TS of the placebo- and API-containing tablets. A scatterplot matrix (**a**) shows the relationships of TS values between placebo- and API-containing tablets. Gray stars represent experimental data points, and blue points represent the data predicted from response surfaces. The relationships between the slope of the regression lines and drug load (**b**).

**Figure 5 pharmaceutics-12-00601-f005:**
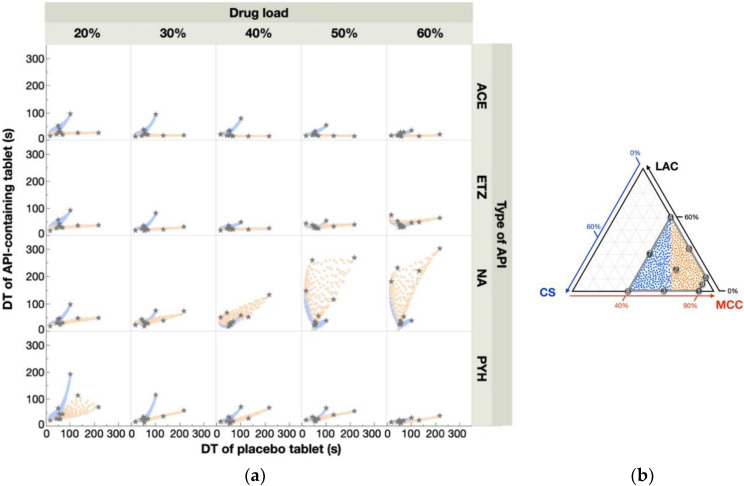
Relationships between the response surface for DT of placebo- and API-containing tablets. A scatterplot matrix (**a**) shows the relationships of DTs between placebo- and API-containing tablets. Gray stars represent experimental data points. Blue and orange points shown in the scatterplots correspond to DTs of untested formulations randomly selected from the left and right halves of the response surface (**b**).

**Figure 6 pharmaceutics-12-00601-f006:**
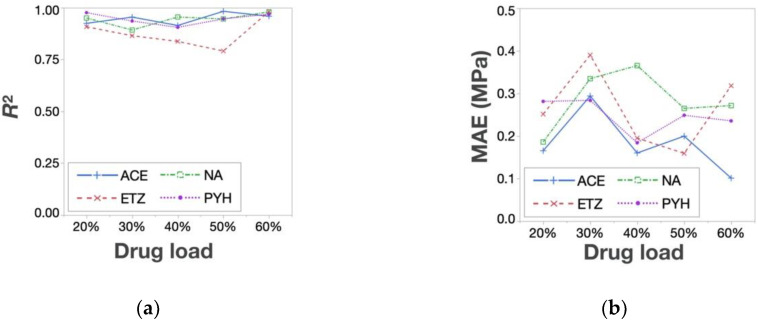
TS Prediction accuracy—(**a**) *R*^2^ and (**b**) mean absolute error (MAE).

**Figure 7 pharmaceutics-12-00601-f007:**
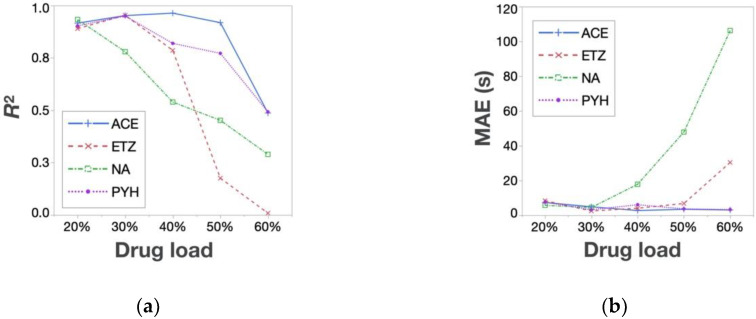
DT prediction accuracy—(**a**) *R*^2^ and (**b**) MAE.

**Table 1 pharmaceutics-12-00601-t001:** Physicochemical properties of model active pharmaceutical ingredients (APIs) and excipients.

Kinds	Component	Median Particle Size Diameter (μm) ^1^	Particle Shape	Solubility ^2^
API	ACE	26.8 ^3^	Nonuniform ^3^	Sparingly soluble in water
	ETZ	7.3 ^3^	Rod-shaped ^3^	Practically insoluble in water
	NA	33.7 ^3^	Nonuniform ^3^	Sparingly soluble in water
	PYH	47.5 ^3^	Square-shaped ^3^	Freely soluble in water
Excipients	LAC	155	Nonuniform	Freely soluble in water
	CS	107	Spherical	Practically insoluble in water
	MCC	66.9	Fibrous and elongated	Practically insoluble in water
	Mg-St	4.9	Nonuniform	Practically insoluble in water

^1^ Particle size analysis was performed with a laser diffraction particle size analyzer equipped with a Tornado dry powder system (LS 13 320; Beckman Coulter, Inc., Brea, CA, USA); ^2^ Data taken from the Japanese Pharmacopoeia, 17th edition, 2016; ^3^ Data quoted from a previous study [22].

**Table 2 pharmaceutics-12-00601-t002:** Excipient composition of model tablets according to an extreme vertices design.

Rp.	LAC (%)	CS (%)	MCC (%)
1	0	10	90
2 ^1^	17.5	17.5	65
2 ^1^	17.5	17.5	65
3	0	35	65
4	60	0	40
5	10	0	90
6	35	0	65
7	30	30	40
8	0	60	40
9	5	5	90

^1^ Rp. 2 is repeated twice because of the central condition.

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
