# Peer review of "A Precise Prediction Method for the Properties of API-Containing Tablets Based on Data from Placebo Tablets"

_pharmaceutics, 2020, doi:10.3390/pharmaceutics12070601_

Round 1
Reviewer 1 Report
General comments
This study complements a first publication published in 2017 which presented a tool for predicting TS and DT parameters using placebo and 4 API formulations at a content of 10%. This work reproduces part of this first study by varying the content of the 4 APIs from 10% to 60%.
In general, the development of a mathematical tool which would make it possible to reduce the number of tests carried out and to predict precisely the formulation which would best meet the desired expectations is a very interesting subject and particularly for the pharmaceutical industry.
However, I wonder about the conditions and especially about the press model used in this work, it is (it seems to me) very far from those used for industrial production.
This raises the question of the possibility of transposing the use of this tool in an industrial context. At a minimum this aspect should be discussed in the publication.
On the other hand, the results obtained highlight a bad correlation for the DT between the experimental and predictive value, the disintegrant present in the formulations is the cornstarch, this disintegrant acts essentially by swelling, it would seem to me interesting to add a disintegrant as crospovidone whose action is more related to wicking. Which would enrich the experiments carried out
For all of these reasons I propose a major revision.
You will find more specific remarks below:
Intro
No data on any other prediction tool?
It seems to me that the concept of QbD should be mentioned in the introduction.
Method
Line 92: 75 ° C for 8H? Why these parameters? Are you sure that the excipients used are stable at this temperature and for this duration?
Line 95: for 12kN, it is preferable to express this data in MPa. You do not justify the choice of this value?
Round Flat?
Line 96: the press used is a hydraulic press, can you specify the production rate of this press? Is this rate compatible with an industrial production rate? what is the compression mode? Top punch? Lower or simultaneous? These parameters can have a significant impact on the characteristics of the tablets obtained and particularly on TS. If the production parameters are far from industrial production parameters this should be addressed in the discussion.
Results and dicussion
Lines 147 to 164: Better define Rp?
Compared to Figure 2, why vary the color scale for the same API
It would seem more visually interesting to take the same scale for the same API?
Reviewer 2 Report
The reviewer would like to congratulate the authors on the extensive work performed for the sake of this study. The study tries to predict the tensile strength and disintegration of time of tablets using information from a placebo formulation. However, in the opinion of the reviewer, the model lacks inclusion of any mechanistic factors that affect TS and DT. In the absence of such an inclusion, the model will severely lack any extrapolative capacity when studying novel APIs not examined in this study. The reviewer highly recommends the inclusion of mechanistic effects to improve the predictive power of the model.
Major comments
- The model is purely statistical and like all statistical models will have poor extrapolative capacity. According to the authors, how can these models be used to predict TS or DT for APIs not included in the study. Furthermore, can the author comment on the predictive power of the model for outside of the studied range?
- Section 3.3, Paragraph 1: Consider simplifying the language and use an example to walk the reader through the application of this method. The text is very wordy and not easy to understand.
- Section 3.3, Paragraph 1: Why are the same drug loads as those used to build the original RS used for validation of the model? This is generally not recommended practice in mathematical modeling.
Minor comments:
- Please indicate what Rp. stands for in the manuscript.
- I suggest that table S1 be included in the main manuscript and not in supplemental information.
- In placebo formulations, is the API replaced by a placebo or does a placebo tablet indicate a tablet with no API and just the three excipients. I believe it is the latter. However, the authors should clarify this in the manuscript.
- Table 1: Please consider providing similar information for excipients.
- Page 3, Line 94: How did the researchers ensure reproducible mixing, especially with the lubricant when hand-mixing the powders in a polyethylene bag?
- Page 5, Line 177: I think the authors would like to say Figure 2.
- Page 7, Line 212: Please provide a figure to show that the slope of the line decreases linearly with increasing API content, that a = -m*(drug load) + c.
Round 2
Reviewer 1 Report
Most of my remarks were taken into account for my specific comments but not the 2 main questions asked in the general comment:
- Problems posed by the use of a hydraulic press?
- Using a different type of disintegrant than cornstarch?
These 2 points were not addressed either in the manuscript or in the authors' responses.
It seems to me that they must at least be addressed in the discussion and possibly mentioned as a perspective to give to this work.
It is for this reason that I recommend a minor revision.
Round 3
Reviewer 1 Report
none